# The Necessity to Investigate In Vivo Fate of Nanoparticle-Loaded Dissolving Microneedles

**DOI:** 10.3390/pharmaceutics16020286

**Published:** 2024-02-17

**Authors:** Ziyao Chang, Yuhuan Wu, Ping Hu, Junhuang Jiang, Guilan Quan, Chuanbin Wu, Xin Pan, Zhengwei Huang

**Affiliations:** 1School of Pharmaceutical Sciences, Sun Yat-Sen University, Guangzhou 510006, China; changzy@mail2.sysu.edu.cn (Z.C.); wuyh83@mail2.sysu.edu.cn (Y.W.); panxin2@mail.sysu.edu.cn (X.P.); 2College of Pharmacy, Jinan University, Guangzhou 511443, China; inzahu@hotmail.com (P.H.); quanguilan@jnu.edu.cn (G.Q.); chuanbin_wu@126.com (C.W.)

**Keywords:** in vivo fate, nanoparticle, dissolving microneedle, necessity

## Abstract

Transdermal drug delivery systems are rapidly gaining prominence and have found widespread application in the treatment of numerous diseases. However, they encounter the challenge of a low transdermal absorption rate. Microneedles can overcome the stratum corneum barrier to enhance the transdermal absorption rate. Among various types of microneedles, nanoparticle-loaded dissolving microneedles (DMNs) present a unique combination of advantages, leveraging the strengths of DMNs (high payload, good mechanical properties, and easy fabrication) and nanocarriers (satisfactory solubilization capacity and a controlled release profile). Consequently, they hold considerable clinical application potential in the precision medicine era. Despite this promise, no nanoparticle-loaded DMN products have been approved thus far. The lack of understanding regarding their in vivo fate represents a critical bottleneck impeding the clinical translation of relevant products. This review aims to elucidate the current research status of the in vivo fate of nanoparticle-loaded DMNs and elaborate the necessity to investigate the in vivo fate of nanoparticle-loaded DMNs from diverse aspects. Furthermore, it offers insights into potential entry points for research into the in vivo fate of nanoparticle-loaded DMNs, aiming to foster further advancements in this field.

## 1. Transdermal Drug Delivery System: A Rising Star

A drug delivery system is defined as a formulation or device used to deliver a specific type of drug (chemical, biological, or natural product) to improve its efficacy and safety [1]. It can be classified according to its chemical nature, physical state, and drug delivery route. Among them, classification according to the drug delivery route aligns closely with clinical medication practices and is widely endorsed by researchers [2]. In this classification, drug delivery systems are typically grouped into categories such as oral, injectable, inhalable, transdermal, and others [3]. These diverse drug delivery systems have found extensive application in clinical settings, benefiting patients suffering from multiple various ailments.

Notably, oral drug delivery systems, injectable drug delivery systems, and inhalation drug delivery systems have insurmountable limitations that restrict their further development. (1) The highly acidic environment of the gastric juice (pH 0.9–1.2) and the digestive enzymes in the digestive tract (e.g., pepsin and trypsin) pose severe challenges to the chemical stability of the drug and the excipients in the drug delivery system; acid/enzyme unstable systems are difficult to develop as oral drug delivery systems [4]. Moreover, the first-pass effect in the liver during oral drug delivery causes biotransformation, which will affect bioavailability [5]. (2) The main problem of injectable drug delivery systems is poor patient compliance. The injection procedure must be performed by healthcare professionals and is mostly accompanied by non-negligible pain, inflammation, and infection at the injection site [5]. This problem is further amplified in the occasion of long-term drug administration, which may cause psychological discomfort and affect the treatment effect. (3) Although the inhalable drug delivery system can achieve better patient compliance, the relevant technology has yet to mature globally. In addition, it is difficult to effectively regulate the deposition site of inhaled drug particles under the current preparation technology, resulting in a relatively low drug delivery efficiency [5].

The transdermal drug delivery system mainly involves delivering drugs painlessly to the blood circulation through the skin [6]. The transdermal drug delivery system has obvious advantages compared to the other three drug delivery systems. Firstly, delivering drugs through the skin can avoid the first-pass effect in the liver. The pH of human skin is nearly neutral, which avoids the pH changes in gastrointestinal transport. Drug absorption is not affected by pH, food, and transit time in the digestive tract, reducing the adverse effects [7]. Secondly, the transdermal drug delivery system exhibits good patient compliance and can be self-administrated [8]. Thirdly, the manufacturing methods of transdermal drug delivery systems are relatively mature. Fourthly, the drug delivery sites are flexible and can be adjusted on demand [9]. Due to these advantages, transdermal drug delivery systems have gained significant interest in the pharmaceutics arena and related fields.

## 2. Challenge of Transdermal Drug Delivery System: Low Transdermal Absorption Rate

With the increase in transdermal drug delivery systems’ clinical applications, physicians, pharmacists, and scientists are concerned about the low transdermal absorption rate, which is commonly found in commercially available products. It is necessary to understand the absorption mechanism to investigate how to improve the transdermal absorption rate. The transdermal absorption rate refers to the proportion of drug molecules that can cross the internal structure of the skin and be absorbed into the blood circulation in a certain period [5]. Studies have revealed that factors influencing transdermal absorption primarily fall into physiological factors, drug properties, and delivery systems [10]. Due to the problem of low drug absorption rate, it is difficult for transdermal drug delivery systems to achieve effective blood concentration, thereby hindering the development of their clinical practice [11].

At the physiological level, skin consists of the epidermis, dermis, hypodermis, and skin appendage [12]. The epidermal pathway is the main one used in drug transdermal absorption [13]. The epidermis includes the stratum corneum, stratum pellucidum, stratum granulosum, stratum spinosum, and stratum basale [14]. It has been reported that the stratum corneum barrier is a crucial factor leading to the low transdermal absorption rate in transdermal drug delivery systems [15]. The stratum corneum is a lipophilic dense layer of dead cells with thickness ranging from 10 to 20 μm [16]. Because of its composition of dead cells and insufficient active uptake ability, only molecules with appropriate lipid solubility and molecular weight can be passively transported across the stratum corneum, and this passive transport process is driven by a concentration gradient with relatively low efficiency [17,18]. Therefore, the presence of the stratum corneum significantly limits the drug absorption rate through the epidermal pathway and the macroscopic transdermal absorption rate.

In addition, drug properties play an important role in transdermal delivery. Specifically, the drug must have certain physicochemical properties to facilitate its absorption through the skin and into the microcirculation. Their properties determine their transport speed within the skin. In general, drugs with low molecular weight, low melting point, and strong pharmacological effects can easily permeate into the skin [19]. In addition, since the stratum corneum has the permeability of a lipid-like membrane and the active epidermis underneath is aqueous tissue, drugs with moderate oil/water partition coefficients have better transdermal permeability and greater penetration. If the drug is insoluble in both oil and water, it struggles to absorb into the skin [20]. Very lipophilic drugs may accumulate in the stratum corneum and cannot be absorbed. For water-soluble drugs, although the transdermal permeability coefficient is small, there may be a high rate of skin penetration when the solubility is high [11].

For delivery systems in recent years, transdermal drug delivery technology has evolved from simplistic methods reliant solely on passive diffusion to sophisticated drug release systems that respond to external stimuli. Transdermal drug delivery has now become a mature technology that can release drugs on demand by controlling the release behaviors [21]. The first generation of the transdermal drug delivery system was used to deliver small molecular, lipophilic, and low-dose drugs. To improve delivering efficacy, the second generation of transdermal drug delivery used techniques such as ultrasound and iontophoresis to deliver the drugs. The third generation of transdermal drug delivery employs technologies such as microdermabrasion, electroporation, thermal ablation, and microneedles to effectively breach the stratum corneum barrier [22].

## 3. Overcoming the Challenge with Dissolving Microneedles

The emergence of microneedle technology has provided potential insights for overcoming the stratum corneum barrier, resulting in the enhancement of the transdermal absorption rate. The microneedle is a chip-type transdermal drug delivery system fabricated by integrating hundreds of needle tips with a length of 25~2000 μm (mostly 100~1500 μm) on a matrix of ~1 cm^2^ area [23]. It can create a large number of tiny pores within the skin as delivery channels to promote the penetration of drugs through physical penetration with the needles [24]. Thus, they can greatly increase the transdermal absorption rate compared with transdermal drug formulations [25]. In addition, delivering drugs through microneedles can achieve a rapid transdermal absorption rate and high bioavailability, which is comparable to subcutaneous injection; the former may even exceed the latter when delivering protein or other biomacromolecules [26]. Furthermore, it can be applied to most of the body’s skin positions [27] and improve patient compliance. It is worth mentioning that during subcutaneous injection, the tip of the syringe always inevitably touches the nociceptive nerves that perceive pain in the skin, resulting in severe pain and low patient compliance [28]. However, by regulating the length of microneedles, it is possible to avoid touching the nociceptive nerves distributed in the deeper dermis and subcutaneous tissues, which improves patient compliance by mitigating pain [29].

Over the past few years, microneedles have evolved with significant development, and the current microneedles can be categorized into solid microneedles, coated microneedles, hollow microneedles, dissolving microneedles (DMNs), and swellable microneedles [30] (Figure 1). (1) Solid microneedles: based on the principle of “poke and patch”, firstly, the metal, ceramic, or silicone-based microneedles without loading drugs puncture the stratum corneum and form delivery channels. Then, the drug formulations will be administrated after the microneedle is removed so that the drugs can diffuse through the delivery channels [31]. However, during the actual drug delivery process, the delivery channels generated by solid microneedles are temporary and easy to close, affecting the transdermal absorption rate. (2) Coated microneedles: based on the principle of “coat and poke”, the drugs are coated at the tip of the needles in the form of membranes; then, the drugs dissolve from the surface of the needle tip after administration [32]. This type of microneedle can only be used for drugs with high efficacy per unit dose and cannot be used for high-dose delivery. (3) Hollow microneedles: based on the principle of “poke and flow”, the drug solution’s flow into the skin is facilitated by the driving pressure after applying microneedles [33]. The hollow structure of this kind of microneedle has poor mechanical properties and is prone to fracture during drug delivery. (4) DMNs: based on the principle of “poke and release”, the drugs are loaded in the tip of the needle, and the tip will be degraded or dissolved, releasing the drugs [34]. (5) Swellable microneedles: based on the principle of “poke and swell”, the gel materials in the microneedle dissolve and swell after administration and then squeeze the skin to form a pore, which leads to drug diffusion [35]. However, this type of microneedle has a complicated preparation process and it is difficult to regulate the drug release behavior.

Compared with the other microneedles, DMNs display several advantages: (1) The needle tips exhibit an occupancy effect, where the formed delivery channels will not close before the needle tips are completely dissolved. Unlike DMNs, solid microneedles do not utilize such an effect, which causes a decrease in the transdermal absorption rate [31]. (2) The whole parts of needles can be loaded with drugs, and therefore, the drug-carrying capacity is higher than that of the coated microneedles [36]. (3) The solid structures of the needles have better mechanical properties than hollow microneedles. Hence, they do not fracture easily in the clinical drug delivery process [33]. (4) The preparation process of DMNs is relatively straightforward compared with swellable microneedles, and the drug release behavior can be regulated more easily [33]. In addition, DMNs are prepared with biodegradable and water-soluble materials, which can avoid the generation of harmful medical wastes [37].

Based on the above-mentioned advantages, DMNs are expected to be a plausible technique to overcome the challenge of low transdermal absorption rates. Moreover, they show great potential for accelerating the development of the global transdermal drug delivery system markets.

## 4. Nanoparticle-Loaded DMNs: A Better Choice for Precision Medication

In contemporary medicine, there is a discernible shift towards precision medicine. This evolving paradigm emphasizes the individualization, refinement, and intelligence of clinical treatments, thus setting forth a new and heightened demand within the medical field.

Although DMNs exhibit several advantages, some unmet needs for precision medicine remain. Specifically, most of the materials used in DMNs show strong hydrophilicity, such as hyaluronic acid (HA) [18], polyvinyl alcohol (PVA) [38], and povidone K30 (PVP K30) [39], while about 75% of marketed products and 90% of drugs in development pipelines are hydrophobic [40]. Therefore, hydrophobic drugs display low solubility in hydrophilic microneedle materials [41], which is attributed to a poor drug loading capacity. In addition, the drug release profiles cannot be controlled by merely using hydrophilic materials, and DMNs consisting of such materials cannot achieve precise and intelligent treatments [42]. Therefore, these two main challenges must be addressed when developing DMNs to fulfill the requirements of precision medicine.

Applying nanocarriers is a novel approach to improve the drug loading and controlled-release capacities of DMNs. Nanocarriers are drug delivery carriers with a diameter ranging from 1 to 1000 nm (mostly 100 to 500 nm) in single or multiple dimensions [43]. On one hand, nanocarriers can enhance the drug loading capacity of DMNs, and the proposed mechanisms are provided below. Firstly, the hydrophobic regions of the matrix materials serve as suitable accommodation media for hydrophobic drugs to improve the drug solubility of nanocarriers [44]. Secondly, at the microscopic level, nanocarriers have a subtle “lattice-like” structure in which the cavities of the “lattice” can accommodate drug monomers or oligomers, thereby facilitating drug dispersion [45]. Moreover, at the nanoscale, the solubility of the drugs will increase significantly as the size of the drugs decreases. The detailed mechanism of this phenomenon can be explained by the Ostwald–Freundlich equation [45], which illustrates that the solubility of a drug is related to its particle size:(1)$$\ln\frac{S_{2}}{S_{1}}=\frac{2\sigma M}{\rho RT}\left(\frac{1}{r_{2}}-\frac{1}{r_{1}}\right)$$
where *r*_1_ is the radius of drug particle 1, *r*_2_ is the radius of drug particle 2, *S*_1_ is for the solubility of drug particle 1, *S*_2_ is the solubility of drug particle 2, *σ* is the surface energy of the drug particles, *M* is the molecular weight of the drug, *ρ* is the density of the drug particles, *R* is the ideal gas constant, and *T* is the absolute temperature. According to Equation (1), the smaller the particle size of the drug molecules, the higher the solubility. Meanwhile, due to the domain-limiting effect of nanocarriers, the distribution diameter of the drug is within the nanoscale range [46]. Therefore, the solubility of the drugs in the nanocarrier can be effectively enhanced.

On the other hand, nanocarriers can achieve a controlled release profile. Firstly, hydrophobic carrier materials act as retardants, which prolong the drug release cycles and reduce uncontrollable burst releases [47]. Secondly, the physical and chemical properties of nanocarrier materials play an important role in the nanoparticle penetration rate and diffusion range. The drug release behavior of nanomaterials can be modulated by physicochemical modification [48]. Finally, some nanocarriers have active or passive targeting effects for specific organs, which can potentially control drug release at specific sites [49].

In addition, applying nanoparticle-loaded DMNs can improve the bioavailability and retention time of drugs, increasing therapeutic efficacy and reducing the toxicity of some drugs [50]. Furthermore, some nanoparticles were designed to construct smart-responsive MNs, which have become a research hotspot. This type of microneedle can intelligently and dynamically regulate drug release in response to changes in pH, glucose, and enzyme concentrations in the internal microenvironment of the tumor or can be stimulated by external temperature, electric field, or light radiation [51]. Nowadays, nanoparticle-loaded DMNs have been applied to cure diseases through chemotherapy [52], gene therapy [53], immunotherapy [54], photothermal [55], and photodynamic therapies [56]. Some drug formulations for treating specific diseases (e.g., cancer, diabetes, and cardiovascular diseases) are in different stages of clinical trials or have been commercialized [57]. In addition, multiple factors that affect the drug loading and release efficiency of NPs have been extensively studied [58,59].

Based on the above statements, developing composite DMNs with nanocarriers for further clinical application is necessary and significant. Combining DMNs with nanocarriers via enriching nanocarriers at the tips of the DMNs can fabricate a new type of nanoparticle-loaded DMN (Figure 2), which possesses the advantages of both DMNs and nanocarriers such as high drug loading capacity, flexible drug release behavior, good mechanical properties, and simple preparation steps.

Currently, formulation scientists have attempted to apply multiple types of nanocarriers including solid lipid nanoparticles (SLNs), mesoporous nanoparticles, gold nanoparticles, micelles, and polymer nanoparticles into the needles of DMNs to prepare nanoparticle-loaded DMNs (Figure 3). It has been reported that these nanocarriers have been used to treat diseases such as diabetes, infections, cancers, and skin diseases topically and systemically [60]. Some representative examples are summarized in Table 1. Herein, nanoparticle-loaded DMNs have a broad clinical application prospect.

However, nanoparticle-loaded DMNs still face some challenges that hinder their further development and clinical application. The incorporation of nanoparticles into MNs increases the complexity of the MN manufacturing process, and additional manufacturing steps are required. Moreover, the compatibility between the nanoparticles and microneedle materials should be fully investigated to maintain the properties of microneedles such as mechanical strength [81]. In addition, the nanoparticles undergo complex processes in the body. Therefore, the in vivo fate of nanoparticle-loaded DMNs must be investigated comprehensively.

## 5. Necessity of In Vivo Fate Study

The in vivo fate of a formulation refers to all the disposition processes encountered in the body by the components of the formulation, including absorption, distribution, metabolism, and excretion (ADME) at the level of major cells, tissues, organs, and systems [82]. In the DMN formulation, the components include the drugs, the needle materials, and the nanocarriers. In the past few years, researchers have focused mainly on the in vivo fate of drugs [83,84,85]. However, with the continuous development of pharmaceutics, scientists realize that multiple types of excipients entering the body also undergo ADME processes [86]. Therefore, the in vivo fate of the drugs, needle materials, and nanocarriers, constituting the in vivo fate of nanoparticle-loaded DMNs, should be considered during formulation design and development.

### 5.1. Necessity Analysis from Clinical Perspective

Although nanoparticle-loaded DMNs show potential for clinical application, as of December 2023, no related products have been approved for clinical application so far. It is presumed that the lack of in vivo fate studies is the main challenge preventing their clinical translation. After transdermal delivery, nanocarriers may be absorbed via the transdermal pathway or the transdermal appendages pathway, and enzymatic degradation may occur during their transfer across epidermal cells, dermal stroma, and vascular endothelial cells [87]. Although DMNs can break through the stratum corneum barrier, the nanocarriers still face certain transfer processes after release [88]. It may be speculated that nanoparticles would undergo these possible in vivo fates: (1) intact nanoparticles are taken up by skin cells and degraded, so do not enter the blood circulation; (2) nanoparticles are degraded before they are taken up by skin cells, so do not enter the blood circulation; (3) nanoparticles travel across the skin cells but degrade before they are taken up into the blood vessels by the vascular endothelial cells; or (4) intact nanoparticles are taken up by the vascular endothelial cells and enter into the blood circulation (Figure 4). In addition, some properties of the nanocarriers remain unknown, including the transdermal absorption rate, degradation rate, the percentage of complete nanocarrier absorption into the blood and the organs, and the excretion kinetics. These will cause difficulties in prescription, administration, therapeutic monitoring, and pharmacovigilance. We hypothesized that only comprehensive in vivo fate studies can illustrate the in vivo fate of nanoparticle-loaded DMNs. Most importantly, understanding the in vivo fate studies can largely avoid therapeutic failures in clinical applications and potentially promote clinical translation.

Numerous studies have endeavored to explore the in vivo fate of drugs and nanoparticles. The drugs and nanocarriers may undergo dependent or independent in vivo fates. Therefore, understanding the in vivo properties and biodistribution of both drugs and nanocarriers is critical to facilitate the development of nanoparticle-loaded DMNs.

Among all drug modalities, small molecules are usually low-molecular-weight compounds with a defined chemical structure [89]. They exhibit specificity depending on the chemical structures and the targets they interact with in vivo. When small molecules are absorbed in the blood, they are distributed through the body’s circulation to various organs and tissues, where they interact with cellular components. Small molecules tend to have a shorter half-life and are easily metabolized by the body, thus limiting their duration in the body [90]. To explore their in vivo fate, Li et al. developed clearing-assisted tissue click chemistry (CATCH) to optically image covalent drug targets in intact mammalian tissues. This study provided a valuable platform for visualizing the in vivo interactions of small molecules in tissue [91]. Pires et al. constructed a novel approach (pkCSM) that uses graph-based signatures to predict central ADMET [92]. Biologics such as proteins and peptides are macromolecular drugs that display different pharmacokinetics compared with small molecules. The high specificity and strong pharmacological activity of biologics contribute to their application in various diseases [93]. Their immunogenicity, complex structures, and spatial conformation are important factors in their in vivo fates [94]. In addition, some single-stranded nucleic acids are absorbed by the liver and kidneys. Absorption of double-stranded nucleic acids can be more challenging, as their double-stranded structure inhibits the ability of the phosphorothioate backbone to enhance absorption [95]. The pharmacokinetic properties of nucleic acid drugs largely depend on the type of chemical modification of the phosphate backbone and ribose of the nucleic acid, as the chemical modification strategy will have a direct impact on the biostability of the nucleic acid drug and its ability to bind to plasma proteins [95]. For example, phosphorothioate modifications can induce nonspecific binding of nucleic acids to plasma proteins, resulting in beneficial effects on blood clearance, biodistribution, and cellular uptake [96]. Therefore, all of the above factors are worth investigating.

Regarding nanocarriers, a thorough investigation into the myriad factors influencing in vivo fates is imperative. It has been reported that the protein corona is an important factor affecting the interaction between nanoparticles and organisms and the in vivo delivery process. The nature of nanoparticles themselves and the environment of the organism can regulate the in vivo delivery process of nanoparticles by influencing the composition of protein coronas [97]. Therefore, paying attention to protein coronas in the in vivo fate studies is necessary. In addition, the size of the nanoparticles was found to determine their elimination and targeting. Du et al. provided a systematic summary of the size-scale laws governing the transport of nanoparticles in the human body and how size-dependent transport can be used to address the main challenges in translational nanomedicine [98]. Furthermore, the shape, hardness [99], surface charge, and surface hydrophilicity of nanoparticles could influence their clearance and translocation [100], which are also worth investigating.

### 5.2. Necessity Analysis from the Industrial Perspective

Clinical application is the landing point of industrialization, and in vivo fate research is conducive to the industrialization of sophisticated formulations. Firstly, the approval of such formulations by most global regulatory agencies requires clinical trials, and in vivo fate research may provide valuable insights for these trials. Secondly, studying the in vivo fate can effectively save research and development (R&D) spending. Currently, the cost of the existing preclinical research stage is relatively low. It is suggested to increase the investment at the preclinical research stage to avoid some risks for clinical trials in advance and to reduce the trial-and-error cost in the clinical stage as a result. Regarding nanoparticle-loaded DMNs, in-depth in vivo fate studies can be applied in the official preclinical research stage to reduce total R&D costs. Thirdly, it is beneficial for the optimization of formulation. The “quality by design (QbD)” strategy proposed by the U.S. FDA early this century is commonly used for formulations in industrialized R&D [101]. In QbD, the critical quality attribute (CQA) is one of the most important factors. Physicochemical properties such as dissolution and stability are often used as CQAs in conventional formulation industrialization (e.g., industrial transformation of oral drug delivery systems represented by tablets and transdermal drug delivery systems represented by patches). It is rational to some extent because traditional formulations have better in vivo/in vitro correlation (IVIVC), and excellent physicochemical properties usually imply good in vivo pharmacokinetics [102]. Therefore, CQAs can reflect the R&D success rate. However, nanocarrier-loaded DMNs are emerging, as well as other complex formulations, and the mere use of physicochemical properties as a CQA cannot predict clinical performance, which makes it challenging to ensure the success of R&D. For this reason, the development strategies should be modified and optimized by the addition of the in vivo fate investigation as a CQA. For example, the transdermal absorption rate should be used as a CQA for the R&D QbD of nanocarrier-loaded DMNs to promote the industrialization of the potential products.

## 6. Entry Points for In Vivo Fate Studies

As we have demonstrated the necessity to carry out in vivo fate studies on nanoparticle-loaded DMNs, the next task is to investigate some entry points. The in vivo fate studies of nanoparticle-loaded DMNs can be designed according to the following sectors (Figure 5).

(1) Drugs: Multiple types of drug modalities, such as small molecules [103], proteins [104], peptides [105], nucleic acids [106], and cells [107], have been loaded into DMNs. The conventional discipline of pharmacokinetics mainly focuses on the ADME processes and has established a relatively systematic research pattern. The in vivo fate of drugs can be elucidated with the assistance of existing technologies [108].

(2) Nanocarriers: Multiple nanocarriers with different structures could be investigated for the in vivo fates. The fluorescent probe tracing method is a reliable, convenient, and widely used method for in vivo fate studies [109]. Nanocarriers can be labeled with different probes to investigate their in vivo fates. However, conventional fluorescent probes will still emit signals after separating from the labeled parent particles, which will interfere with the results. To solve this problem, environment-responsive probes in response to environmental changes have attracted widespread attention [110]. Such probes can distinguish in-carrier signals from free-probe signals effectively. There are three main categories of environment-responsive fluorescent probes, whose principles are based on fluorescence resonance energy transfer (FRET) [111], aggregation-induced emission (AIE) [112], and aggregation-caused quenching (ACQ) effects [113], respectively. The FRET probes are pairs of fluorescent probe molecules with overlay fluorescence emission spectra that can undergo dipole–dipole interactions [114]. The FRET effect refers to the transfer of radiant energy from one probe molecule in the excited state to another in the ground state upon excitation, which can only occur when the distance between pairs of fluorescent probe molecules is less than 10 nm [115]. When FRET probes are loaded in an intact nanoparticle delivery system, they are restricted by the microstructure of the nanodrug delivery system, and the distance between pairs of probe molecules is limited to produce the FRET effect. Meanwhile, when FRET probes are released from the incomplete nanoparticle delivery system into an aqueous biomatrix, the distance between pairs of probe molecules will be significantly larger and the FRET effect will disappear. The AIE and ACQ probes utilize the AIE and ACQ effects, respectively. AIE probe and ACQ probe molecules have large conjugated structures (aromatic groups), which will undergo aggregation in aqueous biological matrices due to hydrophobic interactions, such as π-π stacking, and form aggregates [116]. The ACQ probes emit fluorescence when encapsulated in nanoparticles. However, after experiencing the degradation or destruction of the nanocarriers, the molecules are released and undergo signal switching or fluorescence quenching due to environmental changes. In contrast, AIE probes show exactly opposite properties. They suffer enhanced fluorescence signals when released from the nanoparticles [117]. Therefore, the status of the nanoparticles in the body can be monitored through the signal changes from these probes.

(3) Microneedle: When the DMNs are inserted into the skin, the hydrophilic microneedle will dissolve in the skin interstitial fluid [118]. Nanoparticles are always loaded into the needle of DMNs. Therefore, nanoparticles will be released into the skin and then transported into circulation when the needles of the DMNs are dissolved [81]. The individual DMN degradation behaviors are mainly governed by the needle shape, length, material, etc. Therefore, the in vivo fate studies of nanoparticle-loaded DMNs can be explored from these aspects. Based on this scale, a series of studies have been performed. For example, alternative lengths of the needle span a wide range (25–2000 μm) [119], and the needle length significantly affects the in vivo fate of nanoparticle-loaded DMNs. This is because the dermal region consists of various cells and matrices that interact with the nanocarriers, and the needle length determines the penetration depth of the formulation in the skin [120]. Specifically, at different depths of the dermis, different types and compositions of cells and matrices are likely to affect the transdermal absorption process, ultimately resulting in different therapeutic effects. Therefore, it is essential to investigate the role of needle length on the transdermal absorption rate. Shi et al. investigated the in vivo fate of nanocarrier-loaded DMNs with different lengths of needles. They found that in the spatial dimension, DMNs showed a length-dependent diffusion depth, while in the temporal dimension, the diffusion rates of DMNs with different lengths (400, 800, and 1200 μm) were similar within 24 h of insertion [121].

In addition, the shapes and materials of microneedles also have an influence on the performance. Li et al. investigated DMNs with different needle geometries, including cone, cone–cylinder, rectangular pyramid, and hexagonal pyramid. The in vivo studies demonstrated that cone MNs exhibited the highest dissolution ratio of 80%, whereas the cone–cylindrical MNs had the lowest dissolution ratio of 40% [122]. Aoyagi et al. systematically examined the impact of tip geometry (tip angle, width) on the mechanical properties of MNs. The results revealed that MNs with low tip angles (15–30°) and thin needle shafts (120 μm) effectively enhanced microneedle insertion for efficient drug delivery [123]. The chemical composition of different needle materials determines their water solubility, swelling, and degradation, affecting the drug release from microneedles [124]. As DMNs typically dissolve entirely within seconds to minutes, they typically manifest a sudden release profile, which is advantageous for scenarios requiring rapid onset of action, such as pain relief. To accelerate the dissolving process, an effervescing agent could be added to the needle materials [125]. However, for vaccines or insulin delivery, a sustained payload release profile is desirable to mitigate side effects and reduce the frequency of MN administration. To achieve prolonged release, needle materials with slower dissolution are preferred [126]. Utilizing chitosan as a needle material enables the sustained release of payloads for up to 28 days [127]. Different needle materials afford DMNs varying dissolving properties, thereby offering flexibility in tailoring release kinetics to specific therapeutic requirements.

(4) Administration: The nanoparticle-loaded DMNs may be used for different administration sites, different diseases, or different animals, and the corresponding in vivo fates can be investigated. Different administration sites are likely to affect the in vivo fate of nanoparticle-loaded DMNs, and the reasons are as follows: Firstly, the thicknesses of the epidermis and dermis are different, which implies variable depths of needle entry into the dermal region after microneedle administration. Secondly, different types and contents of cells and matrices in the dermal layer may interact differently with the nanocarriers. Thirdly, different mechanical strengths of skins contribute different shear forces on the microneedles. Fu et al. explored the influence of nanoparticle-loaded DMNs’ application sites with ACQ probes and demonstrated that the transdermal diffusion rate of nanoparticle-loaded DMNs was positively correlated with skin thickness. Ear skin showed the highest transdermal diffusion rate, followed by abdomen and back skin [128]. In addition to the common target tissue skin, MN can still target other tissues and organs, including the eye [129], mouth [130], heart [131], gastrointestinal tract [132], and tumor [133]. Different target tissues may undergo various in vivo fates.

## 7. Outlook

Many factors may affect the in vivo study and are worth investigating. Based on the components of microneedles, drug types and pharmacokinetics, nanocarriers (size, structure, type), needle (material, composition, shape), fluorescence probe (AIE, ACQ, FRET), administration (animals, disease, site, time), etc., can be investigated.

In addition, many new types of DMNs have been developed and gained more attention. For example, gas-propelled DMNs were fabricated to improve drug transdermal efficiency, which utilizes gas as the driving force for skin permeation [134]. Moreover, to achieve transmucosal sequential delivery of multiple drugs, double-layer DMNs were developed to treat oral mucosa diseases [130]. The in vivo fate of these novel DMNs is also suggested to be investigated, and relevant research will further be conducted. Most importantly, the in vivo fate studies of nanoparticle-loaded DMNs will significantly facilitate the development of clinical practice and industrialization.

In the foreseeable future, it is anticipated that in vivo fate studies of nanoparticle-loaded DMNs will emerge as a focal point of research interest. There is a pressing need for more precise tools and methods to reveal the in vivo fate studies of nanoparticle-loaded DMNs. To address a diverse array of therapeutic requirements and cater to the delivery demands of different types of drugs, multiple types of nanoparticle-loaded DMNs will be constructed with enhanced properties. This trajectory is poised to significantly advance the understanding of in vivo fate studies, thereby setting the stage for future exploration in this domain.

Regarding clinical translation, several imperative considerations come to the forefront. Firstly, essential evaluation criteria are in demand for evaluating the in vivo fate studies of nanoparticle-loaded DMNs. Secondly, achieving large-scale industrialized production of microneedle products is a significant challenge. Thirdly, due attention must be accorded to ensuring the safety and stability of both the MNs and the NPs. Finally, the cost implications pertaining to APIs, excipients, and storage conditions emerge as pivotal factors influencing the clinical application.

## 8. Conclusions

Transdermal drug delivery systems have gained great attention in pharmaceutics. As a novel transdermal drug delivery system, nanoparticle-loaded DMNs possess the advantages of DMNs like high drug loading, good mechanical properties, and easy fabrication. Therefore, they demonstrate outstanding potential in clinical application, particularly precision medication. Nevertheless, as of December 2023, no nanoparticle-loaded DMN products have received official approval from the FDA. It is predicted that the lack of in vivo fate studies is the critical bottleneck issue hampering the clinical translation of relevant products. The in vivo fate studies of nanoparticle-loaded DMNs, which will fulfill the requirements of their clinical application and industry transformation, are necessary.

## Figures and Tables

**Figure 1 pharmaceutics-16-00286-f001:**
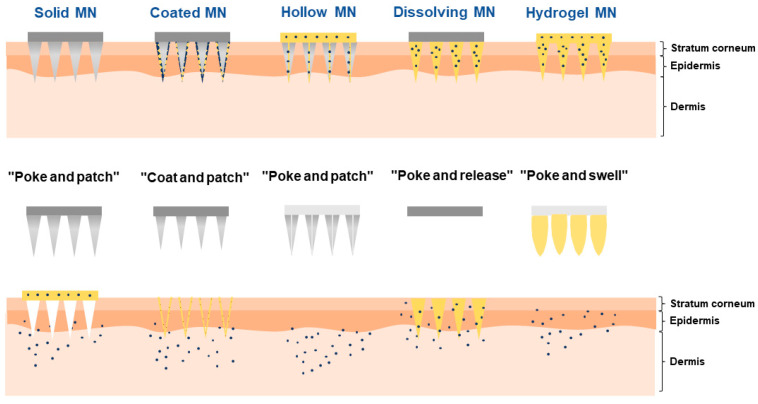
Different types of microneedle.

**Figure 2 pharmaceutics-16-00286-f002:**
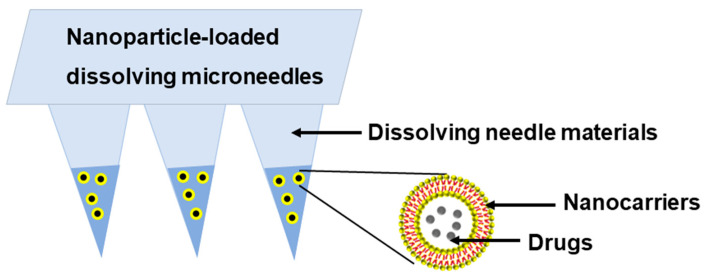
The composition of nanoparticle-loaded DMNs.

**Figure 3 pharmaceutics-16-00286-f003:**
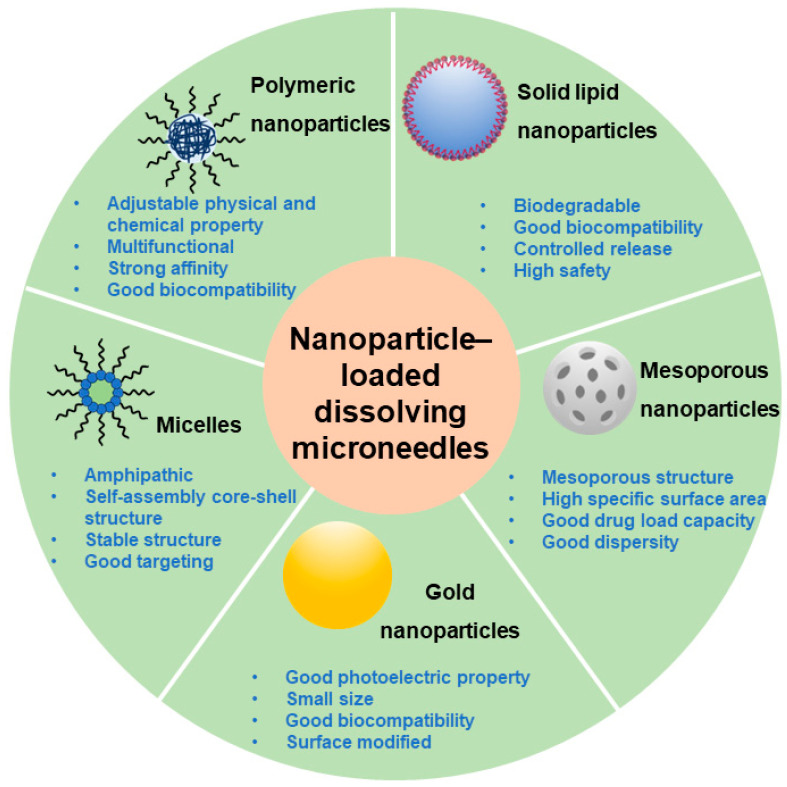
Representative nanoparticles in nanoparticle-loaded DMNs.

**Figure 4 pharmaceutics-16-00286-f004:**
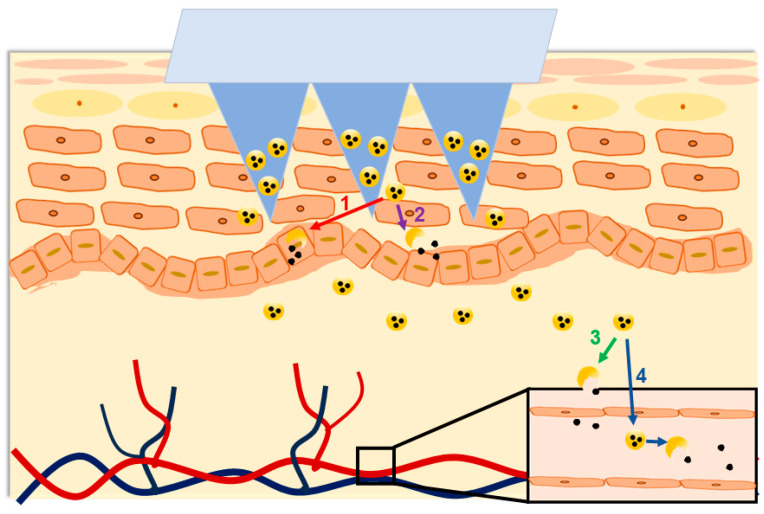
Possible in vivo fates of nanoparticle-loaded DMNs (1 represents that intact nanoparticles are taken up by skin cells and degraded, so do not enter the blood circulation; 2 represents that nanoparticles are degraded before they are taken up by skin cells, so do not enter the blood circulation; 3 represents that nanoparticles travel across the skin cells but degrade before they are taken up into the blood vessels by the vascular endothelial cells; 4 represents that intact nanoparticles are taken up by the vascular endothelial cells and enter into the blood circulation).

**Figure 5 pharmaceutics-16-00286-f005:**
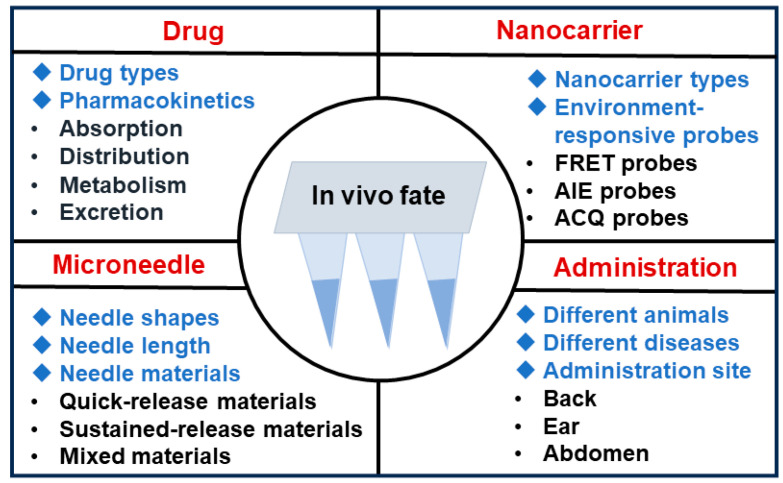
Entry points for in vivo fate studies of nanoparticle-loaded DMNs.

**Table 1 pharmaceutics-16-00286-t001:** Representative studies on nanoparticle-loaded DMNs.

Nanoparticle	Needle Material	Needle Length	Indication	Therapeutic Effects	Reference
Curcumin SLNs	PVA	416 μm	Parkinson’s disease	Enhancing the lipophilicity of curcumin and the treatment of Parkinson’s disease	[61]
Paclitaxel (PTX)/IR-780 SLNs	HA	800 μm	Melanoma	A curable rate of 100% for primary tumor in 30 days	[62]
Doxycycline, diethylcarbamazine, and albendazole sulfone-loaded SLNs	PVA, PVP	500 μm	Lymphatic filariasis	Effective increase in drug concentration in the lymphatic system for the treatment of lymphatic filariasis	[63]
Quercetin and zinc/copper dual-doped mesoporous silica nanoparticles	Sodium hyaluronate, gelatin, sodium alginate	600 μm	Androgenic alopecia	Efficient enhancement of hair growth	[64]
Mesoporous polydopamine nanoparticles loaded with triamcinolone acetonide	HA	700 μm	Oral mucositis	Excellent anti-inflammatory properties	[65]
PEGylated gold nanorod-loaded doxorubicin	HA	600 μm	Human epidermoid cancer	Remarkable antitumor efficacy	[66]
Chloroquine (CQ)/IR780 micelles	HA	800 μm	Melanoma	Effective elimination of primary and distant tumors	[67]
Aggregation-induced emission luminogens (AIEgens) NIR950-loaded polymeric micelles	PVA 103, PVP K30	800 μm	Melanoma	Notable elimination of melanoma tumors with a low dose of NIR950	[68]
PEGylated star-shaped PLGA	Peach gum polysaccharide, PVA, HA	500 μm	Arthritis	Remarkable increase in synovial uptake of Tet and improvement of arthritis	[69]
Polymer nanoparticles loaded with doxycycline (DOX)	PVA, PVP	600 μm	Bacterial biofilm skin infection	Superior antimicrobial and antibiofilm activity	[70]
Polymer nanoparticles loaded with RSV fusion protein	HA, trehalose	520 μm	Respiratory syncytial virus (RSV) vaccine	Induction of robust humoral and cellular immune responses in vivo	[71]
PLGA nanoparticles loaded with influenza matrix 2 (M2) protein antigen	HA, trehalose dihydrate	520 μm	Vaccine	Delivery of antigens into the body with potential for development into vaccines	[72]
PLGA nanoparticles loaded with vitamin D3	PVA, PVP	600 μm	Micronutrient	Excellent delivery of vitamin D3 into the body	[73]
Curcumin (Cur)-loaded micelles	HA, sodium carboxymethyl starch (CMS-Na)	600 μm	Melanoma	High drug delivery efficiency and promising applications in the treatment of melanoma	[74]
Polymer nanoparticles loaded with tetanus toxoid	PVP	250 μm	Vaccine	Highest production of antibodies	[75]
PLGA nanoparticles loaded with dexamethasone (DEX)	Sodium alginate (SA)	500 μm	Inflammatory skin diseases	Excellent delivery of DEX to inflamed areas of the skin	[76]
Polymer nanoparticles loaded with doxorubicin (DOX)	HA	450 μm	Superficial skin tumors	Significant killing effect on superficial epidermal tumor cells	[77]
Poly(caprolactone)(PCL) nanoparticles loaded with Carvacrol (CAR)	PVA, PVP	850 μm/600 μm	Infected wounds	Antimicrobial potential at lower drug concentrations at the site of infection, with considerable antimicrobial activity against Gram-negative bacteria	[78]
PLGA nanoparticles loaded with paclitaxel and indocyanine green	PVA, PVP K30	800 μm	Superficial tumor	Favorable anticancer effect and great potential in cancer treatment	[79]
Micelle nanoparticles loaded with rhodamine B (RhB)	PVA	600 μm	Chronic kidney disease	Raised drug concentrations in the kidneys and excellent preservation of kidney health	[80]

## Data Availability

Details are available from authors.

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
