# Peer review of "The Necessity to Investigate In Vivo Fate of Nanoparticle-Loaded Dissolving Microneedles"

_pharmaceutics, 2024, doi:10.3390/pharmaceutics16020286_

Round 1

Reviewer 1 Report

Comments and Suggestions for Authors

The submitted manuscript review current research on in vivo fate and application of nanoparticle-loaded dissolving microneedles. This is an interesting area and the authors summarized some recent studies and provided some insights. However the writing and content needs to be improved. Some detailed comments:

1.       Many contents were repeated throughout the manuscript and the authors should make sure the key points being delivered efficiently.

2.       The authors need to write more around significance and challenges of nanoparticle-loaded dissolving microneedles.

3.       The figures are oversimplified, which needs to be presented in a more meaningful way.

4.       The information in table 1 is very limited and the authors need to add therapeutic effects if possible.

5.       Conclusion and outlook are very short. The review did not give insights into the future trends and clinical translation. It is better to be included in this review.

Comments on the Quality of English Language

need to be improved

Reviewer 2 Report

Comments and Suggestions for Authors

I suggest to prepare a figure which shows the possible fates of the nanoparticles-loaded dissolving microneedles.

I would suggest to improve the “5. Necessity for in vivo fate study” section of the manuscript since this is the main focus of the review based on the proposed title of the manuscript. Particularly, the “5.1. Necessity analysis from clinical perspective“ should be improved significantly since this can adversely affect health conditions. The authors should elaborate this section giving more examples of already published papers and the possible fate that should be checked based on the chemical structure of the nanoparticles. The author may subdivide this section into 2 sections based on the type of the nanoparticles and type of the drugs.

Comments on the Quality of English Language

Minor editing of English language is required

Reviewer 3 Report

Comments and Suggestions for Authors

In the review entitled “The Necessity to Investigate In vivo Fate of Nanoparticles-loaded Dissolving Microneedles” the authors highlight the need to investigate the fate of nanoparticles-loaded dissolving microneedles by in vivo studies to facilitate the clinical application of such delivery systems.

After briefly describing the main characteristics of drug delivery systems, the authors focus on transdermal drug delivery systems and analyze very superficially the factors involved in drug skin permeation and the properties of different types of nanocarriers.

In this work, the authors cited only few papers that report the use of nanoparticles-loaded dissolving microneedles. In Table 1, the authors cited 9 papers and in section 6 they cited 6 more publications on such topic. The manuscript is mainly focused on the authors’ opinion that in vivo fate of nanoparticles-loaded dissolving microneedles should be thoroughly investigated. Although the authors’ opinion is commendable, obviously, the lack of large number of studies demonstrating usefulness and advantages of nanoparticles-loaded dissolving microneedles does not encourage in-depth in vivo investigations.

In addition, the manuscript contains inaccuracies and contradictions.  For instance, at line 55, the authors report that “the enzyme activity is low, which is attributed to the direct absorption of drugs into the blood circulation after administration and to avoid the first-pass effect”. This statement is not true as the skin has an enzymatic activity and it is able to metabolize different drugs (Kazem, Siamaque et al. “Skin metabolism phase I and phase II enzymes in native and reconstructed human skin: a short review.” Drug discovery today vol. 24,9 (2019): 1899-1910).

At line 73, the statement “transdermal drug delivery systems are mainly used for treating local skin diseases but rarely for systemic diseases” is contradictory. Transdermal drug delivery systems are designed to obtain a drug therapeutic level in the blood and they are not used for topical therapies.

English should be revised.

Comments on the Quality of English Language

English should be revised.

Reviewer 4 Report

Comments and Suggestions for Authors

The manuscript " The Necessity to Investigate In vivo Fate of Nanoparticles-loaded Dissolving Microneedles" by Huang and coworkers deals with microneedles for drug delivery. It is well written and covers an important topic.

Comments:

1. I am a bit puzzled by the manuscript. It reads like a long introduction/motivation for the area. The issue is and the fundamentals are described well. I am missing the description of results from the literature. So the question is: Is it a review article or rather a comment/perspective?

2. I suggest to put more Figures.

3. The fate of drugs and nanocarriers is studied quite a lot and can probably be obtained from these studies analogously. The new area is the combination with microneedles. So interesting is: microneedle fate (degradation, dependence of administration site, interaction of microneedle degradation and nanocarrier release, etc.). To me the only really relevant part of the manuscript is on page 8 parts (3) and (4). The other parts are really just introduction. Thus I suggest to elaborate this more, look for more examples and show more details about the examples mentioned (references 78-82, but also Table 1). There must be studies about the release of nanocarriers from microneedles, which should be covered here.

Comments on the Quality of English Language

The manuscript should be spell and style checked but has a decent quality already.

Round 2

Reviewer 1 Report

Comments and Suggestions for Authors

The authors have well addressed my concerns point-by-point and the quality of revised manuscript is greatly improved. I therefore recommend it be accepted. 

Comments on the Quality of English Language

Can be improved

Reviewer 3 Report

Comments and Suggestions for Authors

The authors revised the manuscript properly.

Comments on the Quality of English Language

English has been improved but minor revisions are still required.

Reviewer 4 Report

Comments and Suggestions for Authors

The revised version addresses all my previous comments.

Comments on the Quality of English Language

I propose to have another spell and style check.